# Risk Factors of Early Atrial Fibrillation Recurrence Following Electrical Cardioversion When Left Ventricular Ejection Fraction Is Preserved

**DOI:** 10.3390/medicina58081053

**Published:** 2022-08-04

**Authors:** Rasa Karaliūtė, Arnoldas Leleika, Ieva Apanavičiūtė, Tomas Kazakevičius, Vaida Mizarienė, Vytautas Zabiela, Aušra Kavoliūnienė, Nijolė Ragaišytė, Daiva Urbonienė, Gintarė Šakalytė

**Affiliations:** 1Laboratory of Behavioural Medicine, Neuroscience Institute, Lithuanian University of Health Sciences, 50009 Kaunas, Lithuania; 2Department of Cardiology, Medical Academy, Lithuanian University of Health Sciences, 50009 Kaunas, Lithuania; 3Kaunas Region Society of Cardiology, 50009 Kaunas, Lithuania; 4Medicine Faculty, Medicine Academy, Lithuanian University of Health Sciences, 50009 Kaunas, Lithuania; 5Institute of Cardiology, Lithuanian University of Health Sciences, 50009 Kaunas, Lithuania; 6Department of Laboratory Medicine, Lithuanian University of Health Sciences, 50009 Kaunas, Lithuania

**Keywords:** atrial fibrillation, cardioversion, left atrial strain, fibrosis, biomarkers, inflammatory, natriuretic peptides

## Abstract

*Background and objectives*: To identify clinical, echocardiographic, and laboratory parameters that affect the early recurrence of atrial fibrillation (AF) after restoring sinus rhythm (SR) by electrical cardioversion (ECV), and to determine whether left atrial (LA) strain, as a noninvasive indicator reflecting fibrosis, is associated with laboratory indicators affecting the development of fibrosis, interleukin 6 (IL-6) or tumor necrosis factor α (TNF-α). *Materials and Methods*: The study included 92 persistent AF patients who underwent elective ECV. The effective maintenance of SR was evaluated after 40 ± 10 days of ECV. Echocardiography, inflammatory markers (high-sensitivity c-reactive protein (hs-CRP), IL-6, and TNF-α), and natriuretic peptides (N-terminal pro b-type natriuretic peptide (NT-proBNP) and N-terminal pro a-type natriuretic peptide (NT-proANP)) were assessed. *Results*: After a 40 ± 10 days observation period, 51 patients (55.4%) were in SR. Patients with AF recurrence had a significantly longer duration of AF (*p* = 0.008) and of arterial hypertension (*p* = 0.035), lower LA ejection fraction (*p* = 0.009), lower LA strain (*p* < 0.0001), higher left ventricular global longitudinal strain (*p* = 0.001), and a higher E/e‘ ratio (*p* < 0.0001). LA strain was an independent predictor of early AF recurrence (OR: 0.65; 95% Cl 0.5–0.9, *p* = 0.004). LA strain < 11.85% predicted AF recurrence with 70% sensitivity and 88% specificity (AUC 0.855, 95% CI 0.77–0.94, *p* < 0.0001). LA strain demonstrated the association with NT-proBNP (r = −0.489, *p* < 0.0001) and NT-proANP (r = −0.378, *p* = 0.002), as well as with hs-CRP (r = −0.243, *p* = 0.04). *Conclusions*: LA strain appeared to be the most accurate predictor of early AF recurrence after ECV in patients with persistent AF. LA strain inversely correlated with NT-proBNP and NT-proANP, but no significant association with any of the inflammatory markers was identified.

## 1. Introduction

Approximately one-quarter of atrial fibrillation (AF) patients are asymptomatic, although others experience a wide range of clinical symptoms which guide the decision between rate or rhythm control strategies [1]. There are several well-known risk factors for AF recurrence, for example, older age, underlying cardiovascular diseases, multiple AF, and left atrium (LA) enlargement [2]. The recurrence of AF is common when selecting a rhythm control strategy: AF recurrence rate after electrical cardioversion (ECV) ranges from 63 to 84% and the success rate of AF ablation in the first year varies from 50 to 80% [3]. A better identification of the most suitable rhythm control strategy candidates is crucial to avoid exposing patients with low chances for long-term sinus rhythm maintenance and to avoid the unnecessary risk of invasive procedures.

Pathophysiologically, atrial fibrosis is associated with the development, progression, and recurrence of AF [4]. The association between AF and atrial fibrosis has been highlighted in several studies [5,6]. LA strain, an easily measurable, noninvasive echocardiographic parameter, has been shown to be inversely correlated with LA fibrosis, evaluated by magnetic resonance imaging in the late gadolinium accumulation phase [7]. Several echocardiographic studies have shown an advantage of LA strain over conventional LA volumetric measurements to predict AF recurrence after ECV and catheter ablation [8,9,10].

Atrial wall inflammation and fibrosis have emerged as important pathogenic contributors in the pathogenesis of AF by altering electrophysiological and structural remodeling [11]. Biomarkers of inflammation and neurohumoral activation as potential predictors of AF recurrence following rhythm control interventions have been investigated by numerous studies [12,13,14]. Elevated natriuretic peptides have been shown to be associated with an increased risk of AF onset and recurrence after intervention [15,16]. Seewöster T. et al. demonstrated that N-terminal pro-atrial natriuretic peptide (NT--pro-ANP) is a strong predictor for the low-voltage areas of the LA and might be considered as a surrogate parameter for atrial myopathy [17]. Several analyses strongly indicate that specific circulating inflammatory markers, such as C-reactive protein (CRP) and interleukin 6 (IL-6), are associated with greater AF recurrence risk after ECV or ablation [18,19,20,21]. However, the results are inconsistent and the use of biomarkers in the management of AF is not recommended by current AF guidelines [22]. Additionally, there is a lack of studies evaluating markers of inflammation and neurohumoral activation as potential risk factors for fibrosis, along with the imaging parameters of LA fibrosis as LA strain.

This study aims to identify clinical, echocardiographic, and laboratory parameters that affect the early recurrence of AF after restoring SR by electrical cardioversion to determine whether LA strain, as a noninvasive indicator reflecting fibrosis, is associated with laboratory indicators affecting the development of fibrosis, IL-6 and TNF-α.

## 2. Materials and Methods

### 2.1. Study Population and Data Collection

Ninety-two patients with persistent AF during the 24-month follow-up period were included and assigned to this prospective cohort study at the Cardiology Department at the Hospital of Lithuanian University of Health Sciences Kaunas Clinics.

We excluded patients with heart failure with preserved or decreased ejection fraction (EF), newly diagnosed severe left ventricular dysfunction (left ventricular EF < 40%), confirmed ischemic heart disease (prior myocardial infarction, coronary bypass surgery, coronary artery stenting or confirmed stenosis ≥ 50%), severe valve disease, an implanted pacemaker, and other known factors which may have affected the results of laboratory tests (creatinine clearance < 50 mL/min, elevation of liver transaminases (AST, ALT) > 3 times the upper limit of normal range, treatment with glucocorticosteroids, active oncological disease, thyroid dysfunction).

The primary outcome was early recurrence (within 40 ± 10 days after intervention), which was assessed according to European Society of Cardiology (ESC) guidelines for AF treatment. Patients were divided into two groups: those who maintained sinus rhythm (SR) and those who relapsed with AF, as diagnosed by clinical manifestations and ECG. The decision regarding medical therapy was made by a physician according to the current European Society of Cardiology (ESC) AF treatment guidelines [22].

This study was carried out according to the Declaration of Helsinki. The study protocol and the written informed consent forms were reviewed and approved by the Kaunas Regional Ethics Committee for Biomedical Research (reference number BE-10-4, date of approval 15 March 2016). Each patient signed an informed consent form that gave the patient enough time to read and ask questions before signing.

### 2.2. Biochemical Analyses

Blood samples for laboratory tests were taken for routine sampling before ECV in the Cardiology Intensive Care Unit of Kaunas Clinics, Lithunian University of Health Sciences Hospital (LUHS), Kaunas, Lithuania, and 40 ±10 days later if SR was maintained, in the Cardiology Clinic’s Department of Consultation and Diagnostics. Blood samples were collected immediately before ECV for all patients, frozen, and were stored at −80 °C until use.

The following laboratory tests were determined: high-sensitivity c-reactive protein (hs-CRP), interleukin 6 (IL-6), tumor necrosis factor α (TNF-α), N-terminal proatrial natriuretic peptide (NT-proANP) and NT-pro type B natriuretic peptide (NT-proBNP).

Serum TNF-α levels were measured using an ELISA kit (DIAsource, Louvain-la-Neuve, Belgium), hs-CRP, COBAS INTEGRA^®®^ 400 plus (Roche-diagnostics, Risch-Rotkreuz, Liechtenstein, Switzerland) and IL-6, a multiplex assay approach using Luminex 100 (Human Cytokine Premixed Multi-Analyte Kit, R&D, Minneapolis, MN, USA). Concentrations of plasma NT-proANP were determined using the human NT-proANP ELISA kit (Biomedica Immunoassay, Biomedica Medizinprodukte GmbH & Co KG, Vienna, Austria) and assessing NT-proBNP levels with the NT-proBNP ELISA kit (PathFast, LSI Medience Corporation, Chiyoda-ku, Japan). The manufacturer’s test characteristics and results were converted from the original unit (nmol for NT- proANP and pg/mL for NT-proBNP) to ng/L in accordance with the quality specifications for type B natriuretic peptide assays and the recommendations of the IFCC Heart Damage Marker Standardization Committee [23].

Serum creatinine levels and creatinine clearance (Cockcroft–Gault equation) were determined before ECV to evaluate renal function [24].

### 2.3. Transthoracic Echocardiography

Two-dimensional echocardiography was performed within 24 h of ECV for all patients using the diagnostic ultrasound system (EPIQ 7, Philips Ultrasound, Inc. Washington, DC, USA) by one investigator. All measurements were calculated over five cardiac cycles with a simultaneously obtained ECG. Two-dimensional measurements were assessed according to the American Society of Echocardiography and the European Association of Cardiovascular Imaging recommendations published in 2016 [25].

The modified Simpson’s disc summation method using apical four- and two-chamber views was used to measure LA volumes. LA maximum volume (Vmax) was measured at the end-systolic frame, just before the opening of the mitral valve. LA minimum volume (Vmin) was measured at the end-diastolic frame, at the mitral valve closure. LA Vmax was indexed to the body surface area to derive the LA volume index (LAVI). LV ejection fraction (EF) was calculated using the modified Simpson’s biplane method, and LA EF was calculated using the following equation (Vmax-Vmin)/Vmax × 100 [26].

Myocardial deformation parameters (left ventricular (LV), global longitudinal strain (GLS) and peak atrial longitudinal strain (LA strain) were obtained by off-line speckle tracking analysis using fully automated 2D speckle tracking software for LV and LA longitudinal strain analysis (Philips QLAB 13.0 software, Amsterdam, Netherlands). The QRS complex was used as the zero-reference point for LA strain analysis.

Mitral peak early (E) diastolic filling velocity was recorded from the apical four-chamber view with 3 mm of PW Doppler sample volume placed between the mitral leaflet tips. Tissue Doppler imaging was performed in the apical four-chamber view with a 5 mm PW Doppler sample volume at the lateral and septal basal regions. The peak septal and lateral early diastolic (e′ wave) mitral annular velocities were collected and E/e′ ratio was calculated. LV diastolic function was evaluated according to the American Society of Echocardiography and the European Association of Cardiovascular Imaging recommendations published in 2016 [27]. The variables for identifying LV diastolic dysfunction and their cut-offs were: septal e’ < 7 cm/s or lateral e’ < 10 cm/s, average E/e’ ratio > 11, LA volume index > 34 mL/m^2^, and peak tricuspid regurgitation velocity > 2.8 m/s. Then, if three variables of LV diastolic dysfunction were positive, the patient was considered as having LV diastolic dysfunction.

### 2.4. Statistical Analysis

Statistical analysis was performed by IBM^®®^ SPSS^®®^ Statistics 20 (IBM Corp., Armonk, NY, USA). The normal distribution of the continuous values was assessed by the Kolmogorov–Smirnov test. Continuous variables are expressed as mean ± standard deviation or median (25th–75th confidence intervals). Categorical variables are expressed as absolute numbers and percentages. Normally distributed continuous variables were compared using the independent sample T-test. If the data were not normally distributed, a Mann–Whitney rank sum test was used. Categorical variables were compared using the Chi-square test. Bivariate correlations between two related variables were calculated using the Pearson‘s or Spearman‘s correlation coefficients (r). Univariate binary logistic regression was performed to identify the risk factors associated with AF recurrence and the results are shown as a hazard ratio (HR) with 95% confidence intervals (CI). Receiver-operating characteristic (ROC) analysis was used to determine the optimal cut-off values with corresponding sensitivity and specificity of continuous variables for the prediction of AF recurrence. A *p*-value < 0.05 was considered significant.

## 3. Results

In total, at 40 ± 10 days of follow-up after the intervention period, 51 patients (55.4%) were in SR. Both groups of participants with AF recurrence and maintained SR were similar in relation to age, sex, medication and cardiovascular risk factors (Table 1).

Compared with patients who maintained SR, patients with AF recurrence had a longer duration of AF from the first episode (7 (2–38) vs. 24 (6–54) months, respectively, *p* = 0.008) and a longer duration of arterial hypertension (5 (4–9) vs. 7 (3–15) years, respectively, *p* = 0.035); however, AF and SR groups did not differ significantly regarding the duration of the current AF episode (*p* = 0.659). Baseline characteristics for the entire patient population are reported in Table 1.

Nearly all patients had LA dilatation, with a mean LA volume index of 50.27 ± 12.76 mL/m^2^, with no difference between AF and SR groups (*p* = 0.104). Patients with AF recurrence had significantly lower LA EF (*p* = 0.009) and LA strain (*p* < 0.0001). AF and SR groups did not differ significantly regarding the LV EF and all participants had a good LV systolic function (EF 54.68 ± 7.92%). Nevertheless, patients who maintained SR showed better baseline LV GLS compared to patients with AF recurrence (−16.57 ± 2.59% vs. −14.58 ± 2.86%, respectively, *p* = 0.001). Patients with AF recurrence and patients who maintained SR did not differ significantly with respect to LV diastolic function (*p* = 0.121). However, patients who maintained SR had significantly lower early-filling velocity–E waves (*p* < 0.0001) and higher values of early-diastolic mitral annular velocity–average e‘ (*p* = 0.026) with respect to a lower E/e‘ ratio (*p* < 0.0001). Table 2 presents the values for all echocardiographic variables between AF and SR groups.

SR group and AF recurrence group showed no significant differences considering inflammatory biomarkers and concentrations of natriuretic peptides (Table 3).

During univariate logistic regression analysis, the following variables were identified as significant predictors for AF recurrence: AF history at first episode > 12 months, LV GLS, E/e′ ratio, LA strain, NT-proBNP > 1335 ng/L, and LA EF (Table 4). To reduce the collinearity problem, E-wave values were not included in the multivariate logistic regression analysis because they were already reported in the derived E/e‘ ratio. In a multivariate logistic regression analysis, only LA strain remained an independent factor for the early prediction of AF recurrence (OR: 0.65; 95% Cl 0.5–0.9, *p* = 0.004) (Table 4).

According to ROC analysis, LA strain < 11.85% could predict AF recurrence with 70% sensitivity and 88% specificity (AUC 0.855, 9 5% CI 0.77–0.94, *p*< 0.0001) (Figure 1).

LA strain demonstrated the association with NT-proBNP (r = −0.489, *p* < 0.0001) (Figure 2a) and NT-proANP (r = −0.378, *p* = 0.002) (Figure 2b), where NT-proBNP correlated more strongly than NT-proANP. Another correlation was detected between hs-CRP and LA strain (r = −0.243, *p* = 0.04), but this correlation was not strong. However, no correlation was found between LA strain and IL-6 (r = 0.014, *p* = 0.916) and TNF-α (r = 0.029, *p* = 0.822).

## 4. Discussion

Our study reaffirms the relevance of the topic, as 44.6% of patients experienced a recurrence of AF. In some studies, the results of AF recurrence after SR recovery were slightly different. Pluymaekers NAHA and colleagues found that, in their study of early and delayed cardioversion, the recurrence of AF was 30% [28]. In another study, the recurrence of AF observed in early (1 month) and late (6 months) periods was 14% and 43% of the subjects, respectively [29]. However, most studies state that after successful ECV, the relapse rate of AF reaches 60% in the first year, which is consistent with our results, taking into consideration a shorter follow-up period of just 40 ± 10 days [26,30,31,32,33,34].

The duration of arterial hypertension and overall AF history were longer in patients with recurrent AF in our study. The significance of clinical factors for recurrence of AF was also analyzed by other researchers. A few researchers have shown that chronic kidney disease, peripheral arterial disease, previously use of beta-blockers, and CHA2DS2-VASc score > 2 points were independent factors for early AF recurrence in the first 30 days after cardioversion [35,36,37,38,39]. Walid Saliba and colleagues emphasized that higher values of CHA2DS2-VASc score allow the direct inference of the occurrence and recurrence of AF [40]. In our study, the distribution of CHA2DS2-VASc score ≥ 2 points did not differ between AF and SR groups.

According to our study, the worse LV systolic function evaluated by LV GLS, worse LA function evaluated by LA EF and LA strain, and worse separate parameters of diastolic function resulted in early AF recurrence after ECV, but we found no LAVI differences between AF and SR groups. In multivariate logistic regression analysis, only LA strain remained an independent predictor of early AF recurrence. Malte Kranert and colleagues reported worse diastolic dysfunction and increased LAVI (>36 mL/m^2^) as a direct and independent predictor of AF recurrence [41]. In another study by Cristina Fornengo and colleagues, it was found that worse LV diastolic function but no LA dimensions or volume significantly correlated with AF recurrence [42]. The importance of LV diastolic dysfunction indicators in predicting AF recurrence following rhythm control has also been confirmed by some other previous studies [40,43,44].

The number of confirmed diastolic dysfunction cases according to the algorithm of the American Society of Echocardiography and the European Association of Cardiovascular Imaging recommendations published in 2016 did not differ between those who experienced an AF recurrence and those who did not have a recurrence of AF in our study. Although LA strain and echocardiographic parameters of diastolic function relate to E wave, the average e‘ and E/e‘ ratios were statistically significantly worse in the AF recurrence group. Some studies have also suggested that LA strain is associated with LA fibrosis, earlier LA geometric remodeling resulting in LA pressure elevation, and may detect diastolic dysfunction earlier [45,46]. Yasada and colleagues found out that LA strain as a single and independent parameter was more accurate in determining diastolic dysfunction compared to the multi-parametric algorithm [47]. Some previous studies have confirmed LA strain as a reliable predictor of AF recurrence after ECV and catheter ablation [48,49,50,51]. The results of our study also confirm the importance of LA strain as a supplementary method to evaluate diastolic function in patients with persistent AF.

Natriuretic peptide levels did not differ between AF and SR groups in our study; however, a univariate regression analysis revealed that higher NT-proBNP levels resulted in a higher risk of AF recurrence. The inverse association between LA strain and NT-proBNP/NT-proANP (the lower the LA strain, the higher the amount of NP) was found. Other studies have shown comparable results: BNP and diastolic dysfunction are independent indicators of AF recurrence [52,53]. Other studies have suggested that the combined assessment of BNP and LA strain is superior in detecting diastolic dysfunction and predicting AF recurrence [54,55]. Some authors have also elucidated the relationship between NP and the LA strain. Mustafa Kurt and colleagues found that NT-proBNP was more associated with LV GLS than LA strain [56]. Thus, we did not find a benefit of NT-proANP in predicting PV recurrence in contrast to NT-proBNP in patients undergoing electrical cardioversion. The benefit of NT-proANP in predicting early AF recurrence remains controversial and has only been demonstrated in certain situations. However, there are a lack of specific studies analyzing the association of NP with LA function in the prediction of AF recurrence and we do not propose to perform natriuretic peptides for the routine diagnosis of AF recurrence.

We observed that inflammatory markers such as TNF-α, hs-CRP, and IL-6 in patients with AF were above normal range. This can be explained by larger LA size, more advanced remodeling, and persistent AF type. These trends are also confirmed by data from other researchers [57]. However, none of the investigated inflammatory markers (hs-CRP, IL-6, TNF-α) differed between AF and SR groups and only a weak association was observed between hs-CRP and LA strain.

Ischaemic heart disease, especially acute coronary syndrome (ACS) and heart failure, are significantly associated with IL-6 and have been implicated as a diagnostic marker by some authors [58]. It seems that higher IL-6 values may mask the true significance of biomarkers in the presence of AF with a patient history of predominantly ACS. Cong Lin-Liu and colleagues showed an inversely proportional association between IL-6 and ischaemic heart disease (both acute and chronic ischaemic syndromes) [59]. Knowing that AF is a significant contributor to mortality in ischaemic heart disease, we noted in the inclusion criteria in the Methods section that patients with ischaemic heart disease were excluded from the study.

The causal relationship between inflammation and AF remains controversial and it is not clear whether inflammation is the cause or consequence of AF [60]. Higher levels of inflammatory cytokines may signal that the remodeling process is more advanced, so it is natural that the rhythm control strategy should be less successful. This is also supported by the data that inflammatory factors statistically differ between AF categories: in paroxysmal AF, the CRP content is lower than persistent AF, but statistically significantly higher than in healthy people [55,56]. Several studies have shown that CRP levels before ECV were higher in individuals who subsequently had recurrence of AF [61,62], but in other studies this difference was not important [63]. There is evidence that plasma IL-6 levels in patients with AF correlate with LA size and AF duration, suggesting that inflammation is associated with atrial remodeling [62]. Currently, there are several small-scale studies that confirm that higher preoperative levels of IL-6 lead to more frequent recurrences in AF, but several studies have not yet found this association [62,64]. The value of inflammatory markers in AF management remains unclear, and thus they cannot be recommended for instigation in daily clinical practice.

## 5. Conclusions

AF history, left ventricular global longitudinal strain, E/e′ ratio, LA ejection fraction, LA strain, and NT-proBNP were identified as significant predictors for sinus rhythm maintenance after electrical cardioversion. LA strain seems to be the best predictor of early AF recurrence after electrical cardioversion in patients with persistent AF. LA strain inversely correlated with NT-proBNP and NT-proANP, but the association with inflammatory indicators was weak or nonsignificant.

## 6. Limitations and Strengths

This study has several limitations. It was a single-center study involving a relatively small number of patients. We chose to perform echocardiography within 24 h after ECV at regular R-R intervals in SR, thus clearly limiting the clinical significance of our findings in daily clinical practice. Our results may be limited by a relatively short observation period of 40 ± 10 days, which may make the results less significant.

The major strength of this study was its strict collection of samples to avoid patients with comorbidities which could affect laboratory and echocardiography results. We aimed to have as homogeneous a sample as possible.

## Figures and Tables

**Figure 1 medicina-58-01053-f001:**
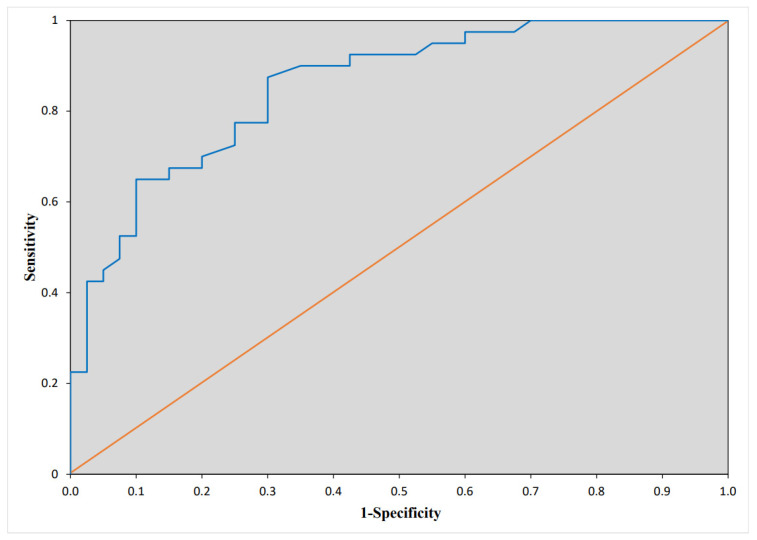
Receiving operator curve of left atrial (LA) strain to predict early atrial fibrillation (AF) recurrence after electrical cardioversion (ECV).

**Figure 2 medicina-58-01053-f002:**
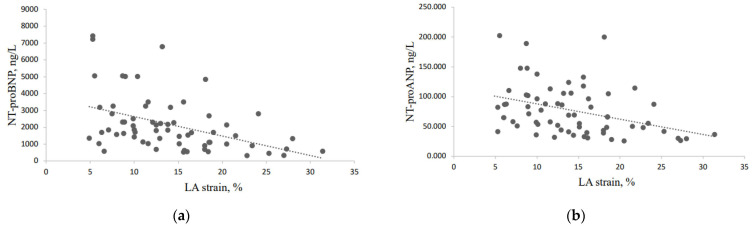
Correlation of left atrial (LA) strain with natriuretic peptides: (**a**) correlation between NT-proBNP and LA strain; (**b**) correlation between NT-proANP and LA strain.

**Table 1 medicina-58-01053-t001:** Demographic characteristics of the subjects.

Risk Factors for AF Recurrence	Total (*n* = 92)	Sinus Rhythm (*n* = 51)	AF Recurrence (*n* = 41)	*p* Value
Age, years	63.30 ± 9.89	61.98 ± 9.24	64.95 ± 10.52	NS
Male gender, *n* (%)	57 (62.0%)	34 (66.7%)	23 (56.1%)	NS
Overweight (BMI ≥ 30 kg/m^2^), *n* (%)	35 (38.0%)	15 (29.4%)	20 (48.8%)	NS
Arterial hypertension, *n* (%)	71 (78.9%)	38 (76.0%)	33 (82.5%)	NS
AH duration, years	5 (4–10)	5 (4–9)	7 (3–15)	0.035
Diabetes mellitus, *n* (%)	13 (14.4%)	6 (12.0%)	7 (17.5%)	NS
Smoking, *n* (%)	21 (22.8%)	9 (19.1%)	12 (31.6%)	NS
Dyslipidaemia, *n* (%)	26 (45.6%)	15 (44.1%)	11 (47.8%)	NS
Duration of AF from the first episode, months	13 (3–47)	7 (2–38)	24 (6–54)	0.008
The duration of persistent				
AF, *n* (%)				
1–3 months	41 (46.6%)	24 (51.1%)	17 (41.5%)	
3–6 months	23 (26.1%)	11 (23.4%)	12 (29.3%)	NS
6–12 months	24 (27.3%)	12 (25.5%)	12 (29.3%)	
First AF episode, *n* (%)	54 (58.7%)	35 (68.6%)	19 (46.3%)	0.036
Beta-blockers, *n* (%)	82 (93.2%)	43 (89.6%)	39 (97.5%)	NS
Class IC antiarrhythmic drugs, *n* (%)	21 (22.8%)	6 (11.8%)	15 (36.6%)	NS
Class III antiarrhythmic drugs (Amiodarone), *n* (%)	42 (45.7%)	27 (52.9%)	15 (36.6%)	NS
ACE inhibitors/ARB, *n* (%)	66 (71.7%)	37 (72.5%)	29 (70.7%)	NS
MRA, *n* (%)	15 (16.3%)	11 (21.6%)	4 (9.8%)	NS
Statins, *n* (%)	18 (19.6%)	11 (21.6%)	7 (17.1%)	NS
CHA_2_DS_2_-VASc score ≥ 2, *n* (%)	65 (70.7%)	36 (70.6%)	29 (70.7%)	NS

Abbreviations: AF—atrial fibrillation; n—number; BMI—body mass index; AH—arterial hypertension; ACE inhibitors/ARB—angiotensin-converting enzyme inhibitors/angiotensin II receptor blockers; MRA—mineralocorticoid receptor antagonists; NS—not significant.

**Table 2 medicina-58-01053-t002:** Echocardiographic characteristics of the subjects.

Echocardiographic Parameters	Total (*n* = 92)	Sinus Rhythm (*n* = 51)	AF Recurrence (*n* = 41)	*p* Value
LVEDD, mm	50.12 ± 3.86	50.29 ± 3.99	49.91 ± 3.75	0.655
MMI, g/m^2^	101.36 ± 22.04	98.50 ± 20.27	104.86 ± 23.78	0.187
LV EF, %	54.68 ± 7.92	54.43 ± 6.55	54.96 ± 9.29	0.767
LV GLS, %	−15.60 ± 2.89	−16.57 ± 2.59	−14.58 ± 2.86	0.001
LA diameter, mm	48.20 ± 4.83	47.61 ± 5.54	48.97 ± 3.63	0.208
LA volume index, mL/m^2^	50.27 ± 12.76	48.24 ± 14.41	52.96 ± 9.74	0.104
LA EF, %	27.95 ± 9.80	30.29 ± 11.03	24.85 ± 6.89	0.009
LA strain, %	14.07 ± 5.88	17.49 ± 5.51	10.65 ± 3.96	<0.0001
E wave, m/s	94.69 ± 17.58	88.08 ± 16.17	103.33 ± 15.62	<0.0001
e‘ average, cm/s	10.28 ± 1.26	10.55 ± 1.28	9.96 ± 1.17	0.026
E/e‘	9.39 ± 2.09	8.53 ± 1.98	10.47 ± 1.70	<0.0001
DT, ms	175.85 ± 20.24	178.10 ± 22.02	173.00 ± 17.65	0.275
Diastolic dysfunction (≥3 criteria)	25 (30.9%)	11 (23.9%)	14 (40.0%)	0.121

Abbreviations: AF—atrial fibrillation; LV—left ventricular; EDD—end diastolic diameter; MMI—myocardial mass index; EF—ejection fraction; GLS—global longitudinal strain; LA—left atrial; E—mitral peak early-diastolic-filling velocity; e’—peak early-diastolic mitral annular velocity; DT—deceleration time.

**Table 3 medicina-58-01053-t003:** Biomarker characteristics of the subjects.

Biomarkers	Total (*n* = 92)	Sinus Rhythm (*n* = 51)	AF Recurrence (*n* = 41)	*p* Value
NT-proBNP, ng/L	2148.26 ± 1714.70	1940.33 ± 1675.78	2421.16 ± 1753.31	0.235
NT-proANP, ng/L	75,898.38 ± 40,536.58	76,042.76 ± 41,615.74	75,696.23 ± 39,679.52	0.972
IL-6, pg/mL	9.95 (1.96–152.05)	8.94 (1.96–152.05)	8.76 (2.57–152.05)	0.772
TNF-α, pg/mL	4.36 ± 1.99	3.96 ± 1.45	4.67 ± 2.30	0.136
hs-CRP, pg/mL	1.53 (0.46–4.09)	1.53 (0.47–5.35)	1.53 (0.43- 3.47)	0.487

Abbreviations: AF—atrial fibrillation; NT-proANP—N-terminal pro-atrial natriuretic peptide; NT-proBNP—N-terminal-pro B-type natriuretic peptide; IL-6—interleukin-6; TNF-α—tumour necrosis factor α; hs-CRP—high-sensitivity C-reactive protein.

**Table 4 medicina-58-01053-t004:** Predictors of AF recurrence in the study population.

	Univariate Logistic Regression			Multivariate Logistic Regression		
	OR	95% CI	*p* Value	OR	95% CI	*p* Value
Age, years	1.0	0.9–1.0	0.153			
AF history from first episode > 12 months	2.6	1.1–6.1	0.028			
LV GLS (%)	1.3	1.1–1.6	0.004			
E, cm/s	1.1	1.0–1.1	<0.0001			
E/e′ ratio	1.8	1.3–2.4	<0.0001			
LA strain (%)	0.7	0.6–0.8	<0.0001	0.65	0.5–0.9	0.004
LA EF (%)	0.9	0.9–1.0	0.018			
NT-proBNP > 1335 ng/L	3.5	1.1–10.4	0.026			

Abbreviations: OR—odds ratio; CI—confidence interval; AF—atrial fibrillation; LV GLS—left ventricular global longitudinal strain; E—mitral peak early-diastolic filling velocity; e’—peak early-diastolic mitral annular velocity; LA—left atrial; EF—ejection fraction; NT-proBNP—N-terminal-pro B-type natriuretic peptide.

## Data Availability

Not applicable.

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
