# Peer review of "Risk Factors of Early Atrial Fibrillation Recurrence Following Electrical Cardioversion When Left Ventricular Ejection Fraction Is Preserved"

_medicina, 2022, doi:10.3390/medicina58081053_

Round 1

Reviewer 1 Report

I could cite some points that have already been included in the limitations defined by the authors:

 1/ The size of the sample used

2/ The authors do not specifically describe what the specific effect of the biomarkers would be on the AUC curve and whether or not they would really be an added value in routine clinical use.

3/ Finally, the results (similars) which the article provides has been previously published and the authors could define and reinforce the clinical utility of their results in daily practice.

Reviewer 2 Report

This manuscript describes a cohort study in which the authors enrolled patients who accepted sinus rhythm restoration from persistent atrial fibrillation (AF) via electrical cardioversion (ECV), and analyzed the clinical data, finding the patients with lower left atrial (LA) strain were associated with higher risk of early AF recurrence. The idea of using LA strain as a predictor of early AF recurrence is pretty new. And the design of choosing participants with sinus rhythm restoration via ECV, plus long-term follow-up, is decent. I listed several major concerns need to be addressed. 

1.    The title: “risk factors for recurrence of early atrial fibrillation…” should be altered to “risk factors for early recurrence of atrial fibrillation…”.

In abstract part, show the full names of EF, GLS for their first appearance. 

2.    In the method part, please state the medication use of the patients (besides beta-blocker and antiarrhythmic drugs) and mention the methods how you measured IL-6, TNF-a and hs-CRP. 

3.    Coronary heart disease (CHD), especially acute coronary syndrome (ACS) is significantly associated with acquired AF (PMID: 28818746). But the CHD status was not mentioned in this paper. How was the CHD and ACS history of the enrolled patients? IL-6 was highly released in ACS and it’s involved in the pathogenesis of both CHD and AF (PMID: 23906495, 29324263). It is likely that IL-6 level in AF patients was already higher than the baseline (health individuals). The high IL-6 level involved with CHD can also mask the real IL-6 change in afterward atrial fibrillation. This can explain why IL-6 showed no difference in AF recurrence. For the readers’ benefit and interest, please discuss more about these questions in the discussion part.

Reviewer 3 Report

The authors have focused on identifying clinical, echocardiographic and laboratory parameters as predictors of early recurrence of atrial fibrillation after electrical cardioversion to sinus rhythm and in this context the value of LA strain and its correlations with various laboratory parameters affecting the development of LA fibrosis. I have the following comments/suggestions:

Keywords should include terms as atrial fibrillation, cardioversion

In the section 2. Materials and methods line 136, you describe as cut-off value for identifying LV diastolic dysfunction an average E/e ratio >11, please explain why since in the recommendations for the evaluation of left ventricular diastolic function by echocardiography (from the cited reference 28) the recommended average E/e ratio is > 14.

In the section 3. Results

1.     In Table 1, is p value identical in all three groups classified according to the duration of persistent AF?

2.     Do you have data about other clinical features that can influence AF recurrence (coexisting COPD or obstructive sleep apnoea, renal disease, vascular disease ….?

3.     Please explain what is meaning beta-blockers, months?

4.     Provide data separately regarding antiarrhythmic therapy with class IC and with class III drugs

5.     Please provide data regarding non-antiarrhythmic drugs as adjunctive therapy for comparative evaluation in the two groups since various drugs (ACE, ARB, MRA, statins.) may have a potential role in preventing AF recurrence. In this regard, I suggest checking and referring to Gheorghe G., et al. Cardiovascular Risk and Statin Therapy Considerations in WomenDiagnostics 2020, 10(7), 483. https://doi.org/10.3390/diagnostics10070483 and Toma, M.M.; et al. Use of anticoagulant drugs in patients with atrial fibrillation. Does adherence to therapy have a prognostic impact?. Biomed. Pharmacother. 2022, 150, 113002. https://doi.org/10.1016/j.biopha.2022.113002

Line 171 “Although patients…” needs reformulation of the sentence, also the English language needs revision, 

Table 2. LA volume is measured in mL/m2? is this indexed LA volume??, please correct

Line 191. The authors identified significant predictors for SR maintenance (table 4); but the title of table 4 is prediction of AF recurrence, there should be concordance between the terms described.

Section 4. Discussion

Line 272 No association with inflammatory markers with LA strain was observed in the study does not match with the results presented by the authors at line 207 where it is described the correlation between LA strain and hs-CRP (p=0.04). Please explain.

After L288. For a comprehensive approach to the topic, as the last paragraph of Discussion section, it is advisable to present the main limitations and strengths of your study.

Conclusions must be reshaped. No needed numerical data, as they were already provided in the manuscript. It must be a Single paragraph, describing the main findings of your work and the novelty/special aspects your paper brings to the field.

I have also two questions for the authors

1.     LA strain was found to be the best predictor of AF recurrence as may be able to indicate LA fibrosis. Where any patients in the study evaluated with cardiac MRI for assessment of LA fibrosis? 

2.     Do you have any explanation for the absence of correlation between AF recurrence and LA volume in this study?

References must be written in the MDPI style, providing all info requested for references. Please see the Instructions for authors – they are given to be respected, not being optional.

Round 2

Reviewer 2 Report

This manuscript describes a cohort study in which the authors enrolled patients who accepted sinus rhythm restoration from persistent atrial fibrillation (AF) via electrical cardioversion (ECV), and analyzed the clinical data, finding the patients with lower left atrial (LA) strain were associated with higher risk of early AF recurrence. The idea of using LA strain as a predictor of early AF recurrence is pretty new. And the design of choosing participants with sinus rhythm restoration via ECV, plus long-term follow-up, is decent. The authors responded to my questions very well and made thoughtful revisions. I listed several concerns need to be addressed. 

1.    Three-line table format is widely used in scientific journals. It is suggested to be applied to all the tables in this paper. 

2.    For the readers’ convenience, please mention this research as a cohort study in line 78.

Author Response

RESPONSE TO REVIEWER

This manuscript describes a cohort study in which the authors enrolled patients who accepted sinus rhythm restoration from persistent atrial fibrillation (AF) via electrical cardioversion (ECV), and analyzed the clinical data, finding the patients with lower left atrial (LA) strain were associated with higher risk of early AF recurrence. The idea of using LA strain as a predictor of early AF recurrence is pretty new. And the design of choosing participants with sinus rhythm restoration via ECV, plus long-term follow-up, is decent. The authors responded to my questions very well and made thoughtful revisions. I listed several concerns need to be addressed. 

  1. Three-line table format is widely used in scientific journals. It is suggested to be applied to all the tables in this paper.

AL: Tables have been converted to a three-line format.

  1. For the readers’ convenience, please mention this research as a cohort study in line 78.

AL: Thank you for your feedback. This is definitely a cohort study, so we are happy to correct and highlight this to the readers.

Reviewer 3 Report

The authors responded to my requests.

Author Response

RESPONSE TO REVIEWER

AL: Thank you for your comments and the opportunity to correct the article.